# Determination of the Fractal Dimension of the Fracture Network System Using Image Processing Technique

**Rouhollah Basirat, Kamran Goshtasbi *** **and Morteza Ahmadi**

Rock Mechanics Engineering Division, Faculty of Engineering, Tarbiat Modares University,
14115-111 Tehran, Iran; r.basirat@modares.ac.ir (R.B.); moahmadi@modares.ac.ir (M.A.)

**\*** Correspondence: goshtasb@modares.ac.ir

**Abstract:** Fractal dimension (FD) is a critical parameter in the characterization of a rock fracture network system. This parameter represents the distribution pattern of fractures in rock media. Moreover, it can be used for the modeling of fracture networks when the spatial distribution of fractures is described by the distribution of power law. The main objective of this research is to propose an automatic method to determine the rock mass FD in MATLAB using digital image processing techniques. This method not only accelerates analysis and reduces human error, but also eliminates the access limitation to a rock face. In the proposed method, the intensity of image brightness is corrected using the histogram equalization process and applying smoothing filters to the image followed by revealing the edges using the Canny edge detection algorithm. In the next step, FD is calculated in the program using the box-counting method, which is applied randomly to the pixels detected as fractures. This algorithm was implemented in different geological images to calculate their FDs. The FD of the images was determined using a simple Canny edge detection algorithm, a manual calculation method, and an indirect approach based on spectral decay rate. The results showed that the proposed method is a reliable and fast approach for calculating FD in fractured geological media.

**Keywords:** fractal dimension; fracture network; image processing; rock mass

---

## 1. Introduction

Manual/conventional measuring of the geometry of rock mass discontinuities is often a slow and tedious procedure. In many cases, a rock outcrop is mainly inaccessible, which makes it difficult to determine such a geometry. Therefore, it is necessary to achieve a fast and reliable method for the extraction of geometric parameters of discontinuities. Digital image processing techniques (DIPTs) have high efficiency in this field. The advantages of these methods are higher levels of safety, quickness, and accessibility to all joints and less error in data collection compared to manual methods. In this method, rock mass discontinuities are identified and some geometric parameters of discontinuities are obtained based on the differences between the gray level of the discontinuities and the intact rock.

The early applications of digital images in geoscience date back to 1976, when McCarter [1] applied photographs of a rocky open-pit mine to determine the position of geological structures. Reid and Harrison [2] used a series of DIPTs to determine the discontinuity of a rock surface. Kemenya [3] presented a 3D model of fractures using the 2D image processing method of discontinuities. Otoo et al. [4] combined typical images with laser images and DIPTs in order to obtain the 3D properties of a fracture. Hong et al. [5] applied DIPTs to estimate the geological strength index (GSI). Mohebbi et al. [6] used DIPTs along with fuzzy logic to determine the system of discontinuities in the Choghart mine.

Obtaining information on a rock mass fractures network system (FNS) is an important requirement for an optimal design in rock engineering and mechanical behavior of a rock mass. One of these parameters is the fractal dimension (FD) of the FNS. Fractalness means that details can be seen regardless of the scale of observation. A very distinct feature of fractal phenomena is the scale invariance. This feature is another important attribute in fractals. If a given part of the fractal phenomenon is magnified, the enlarged portion will have a similar geometry to the original state of the phenomenon. By repeating this process in the selected part, the shape of the result is again the same as in the previous section. This process, which can be continued infinitely, is known as "self-similarity". In nature, many objects and events are self-similar, and geological structures are no exception to this rule [7–12].

Since the presentation of fractal geometry by Mandelbrot [13], its application has also been initiated in other sciences. Some researchers have used fractal analysis to investigate the role of geological features in controlling the mineral deposit occurrence inferred from the analysis of their spatial pattern [14–16]. In recent years, many studies have been conducted to understand the existing regularity in the structural geology processes [8,12,17]. According to these studies, tectonic processes can be described by fractal concepts [17]. For example, many features of geological structures can be identified by calculating the FD of linear structures, such as faults and waterways. It is also possible to determine and compare the density of fractures in a specific range [17].

Although a comprehensive definition that can be included in all of these applications has not yet been provided, a fractal system can be defined by Equation (1) [17]:

$$N_i = C / r_i^D \tag{1}$$

where $N_i$ is the number of objects (fragments characterized by the linear dimension $r_i$), $C$ is proportionally constant, and $D$ is FD, which is calculated by Equation (2):

$$D = \frac{\log\left(\frac{N_i+1}{N_i}\right)}{\log\left(\frac{r_i}{r_i+1}\right)} \tag{2}$$

In a fractal study, the structural geology phenomenon FD can be determined using the box-counting method (Figure 1). In this method, a square with a length of unity is considered as the base network of fractal structure. The base network is a zero-order square that produces orders higher than itself. In each case, the zero-order square is divided into nine squares at first order, each with $r_1 = \frac{1}{3}$. To create the second-order squares, the first-order squares are divided into nine smaller squares with lengths of one-ninth. This process continues and, in each step, the number of remaining squares is counted as the $N$ index. The FD in this method can be a number between 0 and 2 [17], which its integers are interpreted as follows:

- $D = 0$: Zero represents the Euclidean dimension of a point; this is appropriate, since, as n→∞, the remaining square becomes a point (Figure 1a). In other words, the studied phenomenon has a negligible distribution in the region.
- $D = 1$: When the squares are converted into a line, then Euclidean dimension gives a line (Figure 1c).
- $D = 2$: This value, which represents the Euclidean dimension of a plan, is expected, since all squares are assigned at each level (Figure 1e). In other words, the examined phenomenon has a very high distribution throughout the region.

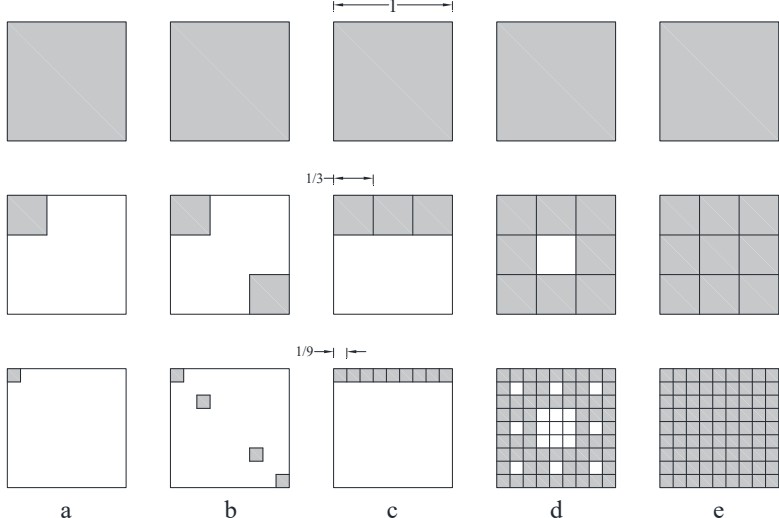

**Figure 1.** The illustration of five 2D fractal structures; (**a**) $N_1 = 1$, $N_2 = 1$, $D = \log 1/\log 3 = 0$, (**b**) $N_1 = 2$, $N_2 = 4$, $D = \log 2/\log 3 = 0.63$, (**c**) $N_1 = 3$, $N_2 = 9$, $D = \log 3/\log 3 = 1$, (**d**) $N_1 = 8$, $N_2 = 64$, $D = \log 8/\log 3 = 1.89$, and (**e**) $N_1 = 9$, $N_2 = 81$, $D = \log 9/\log 3 = 2$.

The concept of a fractal is used for satisfying various purposes in geological problems. One of these applications is the frequency-size distribution of fractures. Under various circumstances, frequency-size distribution of fractures, faults, mineral deposits, oilfields, earthquakes, and topography can be represented by the concept of fractals [8]. Some studies have reported the use of a fractal-based model in the geology to investigate relationships between the properties of pore structure, soil, and FD in different locations, such as tight sandstone and mudstone [18–20]. Gong et al. [21] utilized a FD-based method to describe and predict the complex natural fractures in the tight conglomerate reservoirs. To quantify the fracture characteristics, they calculate fractal dimensions using a box-counting technique and indicate that the spatial distribution of fractures in these reservoirs have fractal features. Chen et al. [22] also used this analytical FD-based technique to evaluate the changes of microstructure and strength of altered granite and their evolution in a certain period. They found a linear relationship between the FD and the rock's mechanical characteristics in symmetrically-shaped tunnels and roadways. Bagde et al. [23] presented FD based on the Bieniawski rock mass rating (RMR) using data obtained from several mines in India. Kulatilake et al. [24] used the factor, FD, to evaluate the visual geology of the tunnel mapping to decide the statistical homogeneity of the rock mass around the tunnel. To this goal, they developed a computer-based program to calculate the FD by the box-counting method. Lei et al. [12] scaled the fracture size of 2 m × 2 m to 54 m × 54 m using the fractal concept. Pavičić et al. [25] obtained the FD factor of three dolomite rock outcrops in Croatia by the box-counting method. The higher value of the FD indicates the more similarity of the fracture's size in the total rock outcrop.

In this work, the FD of the FNS of some geological regions was calculated using the DIPT algorithm based on the improved Canny edge detection method. The FD was also determined using the simple Canny edge detection algorithm, the manual calculation method, and an indirect approach based on spectral decay rate. Finally, the results of these methods were compared and discussed.

## 2. DIPT of Rock Media

An alternative method for the traditional method of surveying discontinuities is the use of DIPTs on the digital images. The gray-scale image is a 2D function that is represented by $f(x,y)$, where $x$ and $y$ are the spatial coordinates of each pixel of the image and the value of $f$ in the coordinates $(x,y)$ is the intensity or the gray surface-height of the image at a particular pixel. The digital image consists of a finite number of elements, each of which having a specific place and a certain number of pixels.

The digital image is a two-dimensional matrix (Figure 2) in which the color of each element (pixel) is represented by a discrete value [26].

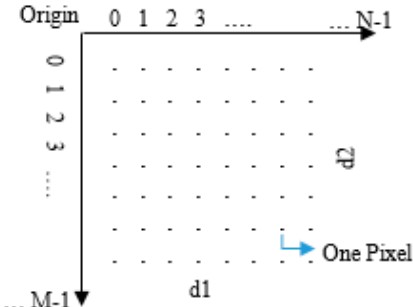

**Figure 2.** Digital image as a 2D matrix [26].

In DIPTs, discontinuities in a picture are described as the boundary between two regions with different gray levels. Changing grayscale in the image is the basis of edge detectors in image processing. This technique provides a quick, complete, and effective way to determine the geometry parameters of the joints by minimizing user interference and overcoming the limitations of the manual method [3].

In the present study, some different fractured geological regions are investigated (Figure 3), where the DIPT is used to compute the FD as a parameter, representing the geometry of the discontinuity system.

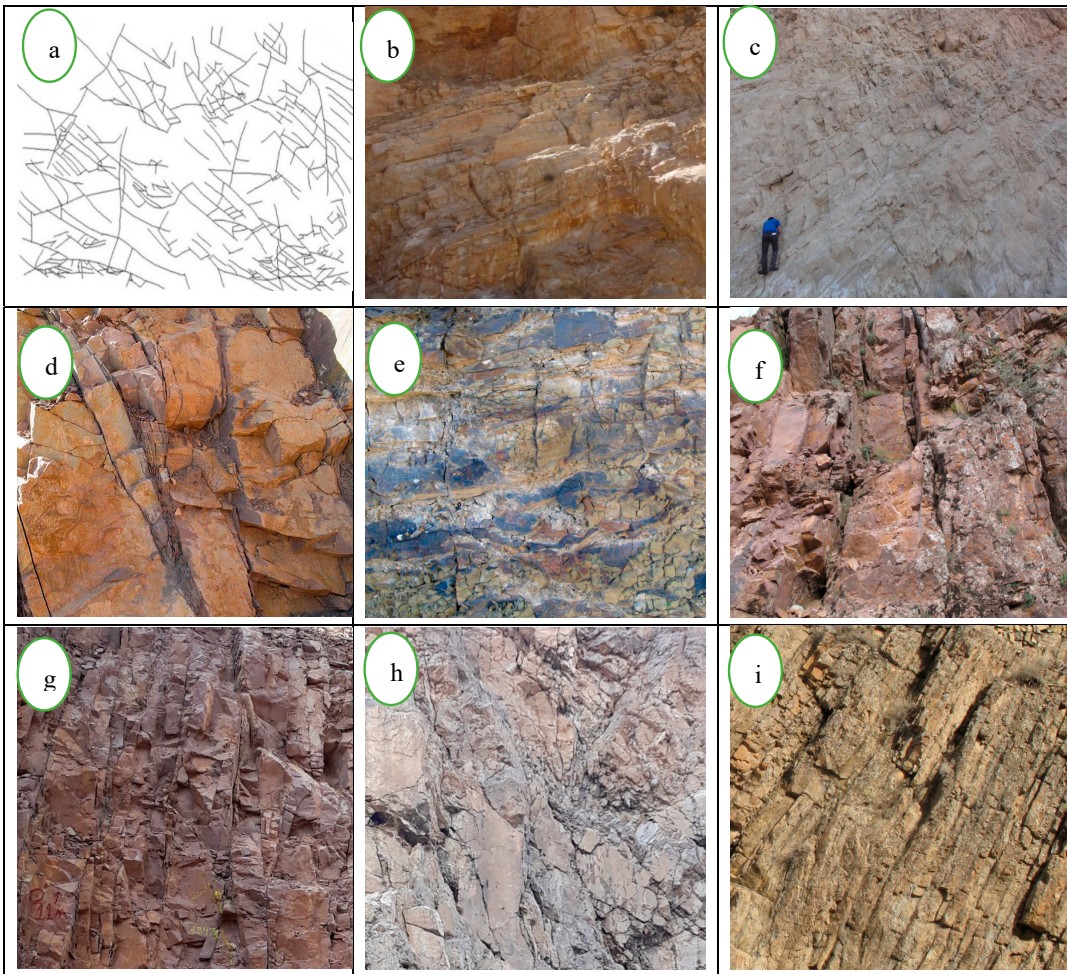

**Figure 3.** Nine examples of fractures network systems (FNS) in geological outcrops.

## 3. The Process of Determining the FD Using DIPT

Figure 4 presents the procedure of FD calculation for one of the rock outcrops (Figure 3c). Digital images of rock outcrops were taken using a 20-megapixel digital camera. The stages of work were as follows:

(a)　Digital image preparation from the face: The brightness of the environment was one of the most important points during the photography. For better results, photography should be done perpendicularly to the face.

(b)　Converting a color image to a monochrome photo: Although the images taken by digital cameras are usually colored, their grayscale-equivalent versions (consisting of a matrix of gray-level pixels) were used for the ease of image processing.

(c)　Pre-processing of digital rock mass photos using histogram equalization: In general, the aim of the pre-processing was to reduce noise and eliminate unwanted details, such as the gap among the lines in the image. Here, the noise refers to any undesirable information in the image [26]. In this research, histogram equalization was employed to adjust image intensities in order to enhance the image's contrast. For more details, refer to Reference [26]. In order to reveal the discontinuities in the image, it was essential to mark and interconnect the represented pixels as discontinuities. These pixels are known as edges in the image [26]. Some algorithms have been proposed for improving these methods by considering some factors, such as image noise and the nature of the edges. The Canny algorithm is one of the best edge detection algorithms. In this algorithm, edge detection was considered as an optimization problem [27]. In this research, we set one optimum threshold for the Canny edge detection algorithm via a trial and error method. Therefore, we utilized the same threshold to perform the edge detection procedure on different examples. More details are presented in the reference [28].

(d)　Calculation of the FD of the FNS.

It is very difficult to determine the fractal dimension by using the definition of fractals. In many cases, it is done by graphical representation. This graphical resolution consists of plotting the function $\log N(r)$ versus $\log(r)$. The resulted curve is linear and its slope is attributed to the FD.

Various fractal analysis techniques can be applied to the study of the spatial distribution of fractures [17]. Among the various techniques, the box-counting technique is used to measure 2D fractal dimensions. In this paper, this method was also used to calculate the FD. This method is highly efficient for the analysis of isotropic and complex 2D patterns [29].

The studied image, including an FN with its edges resulted from the previous steps, is covered by 2D squares in the initial step. However, an appropriate algorithm can be used for more precision to link the pixels of the edges to form the different boundaries of the image and also to remove the unwanted edges. Hough transform is one of the most important methods for identifying and connecting pixels in the form of lines [30]. The main advantage of this transformation is its flexibility in describing the properties of the fractures' boundaries and the ineffectiveness of the noise in the binary image. This method can detect lines divided into segments and can illustrate them continuously in a straight line. After Hough transform grouping, the noise was completely eliminated with the help of opening an operation to improve the results of FN identification. We then performed closing to fill new gaps between small lines. More detailed information is provided in Reference [26].

The next repetition step was then obtained to compute the FD. At this step, the square size was reduced using a reduction coefficient. For each square grid considered in the calculations, a window of the corresponding box-size was created. This window was transmitted from a location to another location in a hypothetical network in the studied image. At every step, the program verified whether or not there was any trace of the investigated feature falling in this box. The number of boxes needed to completely cover the feature was counted and plotted in a log-log graph in terms of the number of squares and their size. Each repetition was placed as a point in this graph. Finally, the slope of this graph was calculated as the FD of the image.

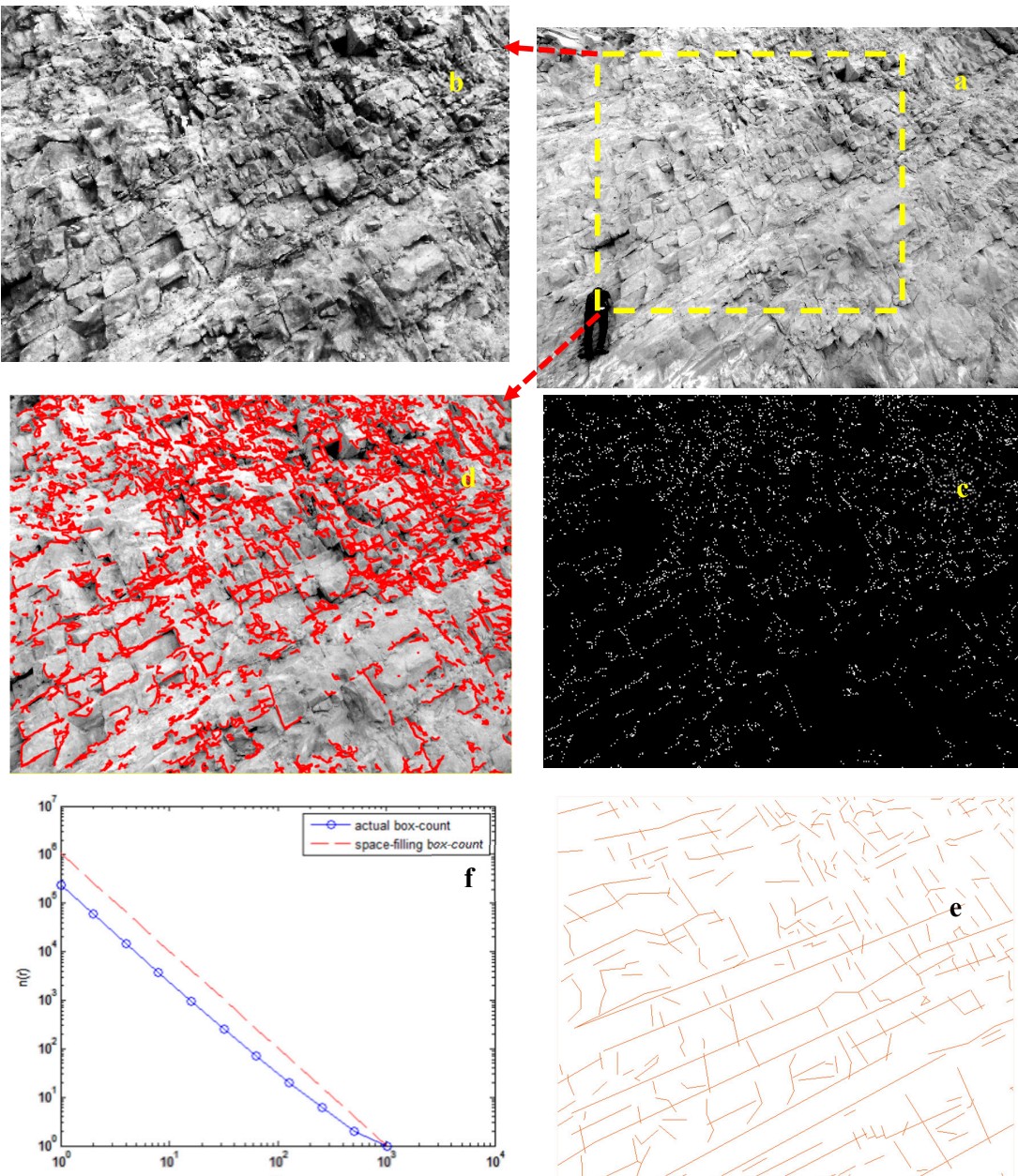

**Figure 4.** Stages of determining the fractal dimension (FD) using the image processing technique: (**a**) Black-and-whitening the original image; (**b**) pre-processing the image in the considered area by the histogram equalization operator; (**c**) detecting the edge points with the Canny algorithm; (**d**) displaying the identified points in the image; (**e**) the fractures finally detected using the Hough transform, and after the required corrections; and (**f**) a log-log diagram in terms of the size of the squares and determining the slope of the line as the FD.

## 4. Results and Discussion

The FD of nine geological medias (Figure 3) was calculated using the proposed method (DIPT). To validate our proposed method for the determination of FD of rock mass images, we performed three other methods based on spectral decay rate; an indirect method, a manual calculation approach, and a simple canny edge detection (without edges improvement via closing procedure).

Bies et al. [31] proposed an indirect method based on spectral decay rate for the FD calculation of n-dimensional data. They presented a generalized relationship among FDs and power spectrum decay-rate parameters (*β*) using empirical validation. To compare the efficiency of this method to our

box-counting and improved canny-based algorithm, we also used this indirect method to calculate the FD. In this regard, we employed a standard method (based on two-dimensional Fourier transform and surface computation) for β extraction from the digital images. FD was then calculated using Equation (3).

$$D = E + \frac{(F + 2 - \beta)}{2} - 1 \tag{3}$$

where *E* and *F* indicate the Euclidean dimension of object and the number of variables of Fourier transform, respectively, and β shows the spectral decay-rate of data.

To calculate the spectral slope, we extracted the spectrum of the images using a two-dimensional fast Fourier transform (FFT). The spectral slope ($\beta$) was then computed via fitting a linear model to the spectra. Finally, the FD was resulted from Equation (3).

As an example, the manual calculation method for Figure 3c, with $r_i = 1/2$, and five iteration steps is shown in Figure 5. After calculating the FD using different mentioned methods, relative errors of three FD calculation algorithms were computed to evaluate the efficiency of three automatic methods. Table 1 shows the percentage of the relative error of these methods for the FD computation of nine images. M1, M2, and M3 indicate simple canny, improved canny (the method presented in this study), and spectral decay rate-based methods, respectively.

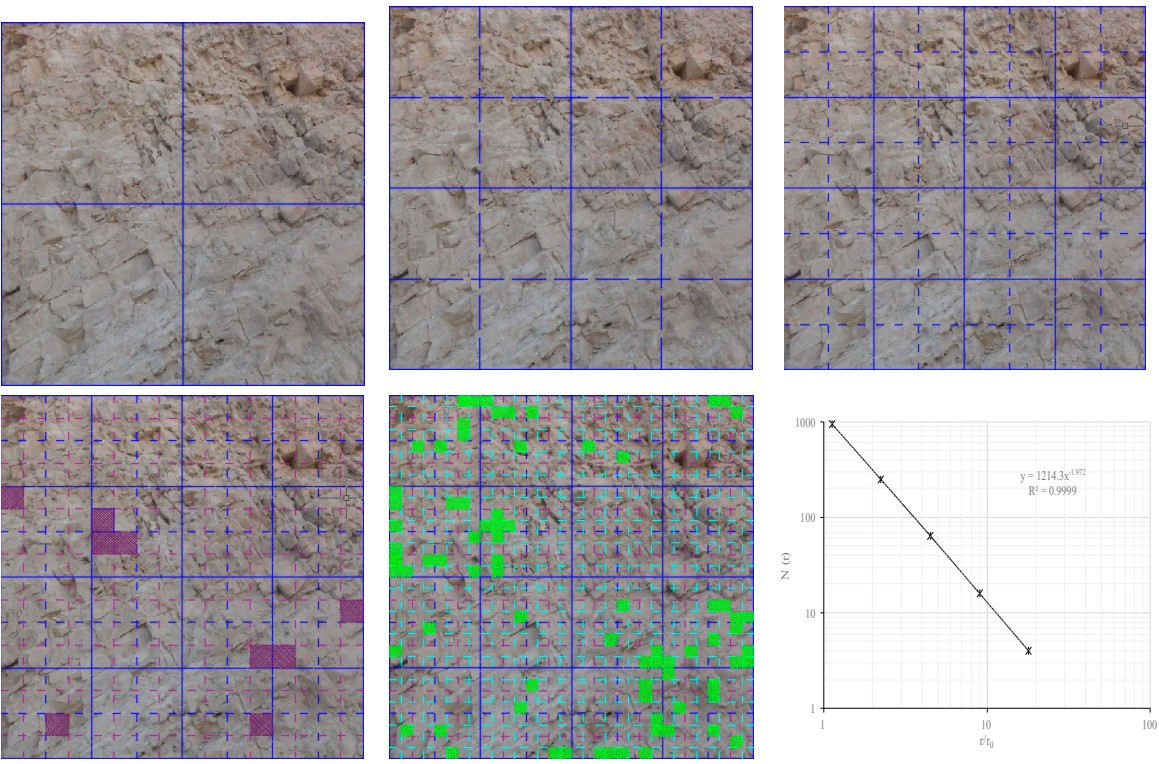

**Figure 5.** Manual method for calculation of the FD in five iteration steps. Fill squares show the empty box (without fracture trace).

Utilizing M1 and M3 methods for FD calculation resulted in 3.023% and 1.694% relative error, respectively; whereas the error of our proposed method was 1.086% for nine images. Therefore, using the improved canny edge detection and box-counting algorithms achieved better performance for FD estimation compared to the two other methods (M1 and M3).

On the other hand, Lee et al. [32] calculated the FD for Figure 3a using the manual method. This simple image was also used for calculating the FD by different mentioned methods. The calculated FD of Figure 3a was equal to 1.65 by the proposed method which showed a good agreement with

the results of Lee et al. [32]. Moreover, the results of our method for other geological images were in meaningful agreement with the reference method (manual method).

**Table 1.** The calculated FD of nine images using different methods and relative errors.

| Image Index | FD: Reference Value | FD: M1 | FD: M2 | FD: M3 | Error M1 (%) | Error M2 (%) | Error M3 (%) |
|---|---|---|---|---|---|---|---|
| a | 1.620 | 1.668 | 1.650 | 1.647 | 2.963 | 1.852 | 1.667 |
| b | 1.801 | 1.763 | 1.821 | 1.857 | 2.110 | 1.110 | 3.109 |
| c | 1.972 | 1.959 | 1.990 | 1.981 | 0.659 | 0.913 | 0.456 |
| d | 1.782 | 1.803 | 1.799 | 1.823 | 1.178 | 0.954 | 2.301 |
| e | 1.890 | 1.972 | 1.943 | 1.943 | 4.339 | 2.804 | 2.804 |
| f | 1.844 | 1.810 | 1.837 | 1.849 | 1.844 | 0.380 | 0.271 |
| g | 1.842 | 1.659 | 1.845 | 1.897 | 9.935 | 0.163 | 2.986 |
| h | 1.796 | 1.831 | 1.812 | 1.781 | 1.949 | 0.891 | 0.835 |
| i | 1.839 | 1.798 | 1.826 | 1.854 | 2.229 | 0.707 | 0.816 |
| Average Relative Error (%) | | | | | 3.023 | 1.086 | 1.694 |

Reference value: Manual method, M1: Simple Canny edge detection, M2: This study, M3: Bies et al. [31].

## 5. Conclusions

This paper proposed an automatic and efficient method in order to determine the rock mass FD using digital image processing techniques. To achieve this aim, the FD was calculated using the box-counting method under MATLAB software. This method was applied randomly to the pixels detected as fractures. In the proposed method, the intensity of the image brightness was corrected using the histogram equalization process and applying the smoothing filters to the image, followed by revealing the edges by the Canny algorithm. This algorithm was implemented in nine fractured rock mass images and their FDs were calculated. The results of this study were compared with three methods, including a manual approach (as a reference method), a simple Canny edge detection method, and an indirect method based on spectral decay rate. The results were in good agreement with the manual method. The results showed that the proposed method was more accurate than simple canny-based and spectral decay rate-based methods. Therefore, this technique is a reliable and fast method for different aspects of engineering geology and rock mechanics.

**Author Contributions:** Validation, R.B., K.G.; investigation, R.B.; writing—original draft preparation, R.B.; writing—review and editing, K.G. and M.A; visualization, R.B.; supervision, K.G.; project administration, K.G. and M.A.

**Funding:** This research received no external funding.

**Conflicts of Interest:** The authors declare no conflict of interest.

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
