# Peer review of "Determination of the Fractal Dimension of the Fracture Network System Using Image Processing Technique"

_fractalfract, doi:10.3390/fractalfract3020017_

Round 1

Reviewer 1 Report

The authors have acted upon my previous comments and added more data to their study. They have also made some additional changes to the writing of the paper. I still wonder about the novelty of the work but I am ok with recommending the paper for publication. 

Reviewer 2 Report

This paper is very interesting and relevant in geological science and other purposes. At this moment, the effort of authors is appreciated, and I have only two questions to ask:

1) Can you describe rock type characteristics in your paper? And why you select this kind 

of rocks?

2) Fractal analysis is divided in singular fractal and multifractal analysis. Singular fractal analysis quantifies the regularity trend in the reproduction of geometric irregularities along scales of clay, silt, and sand domains, with the single fractal dimension being interpreted in the power law dependence. However, in natural systems, PSD does subject to a simple power law. Based on this, multifractal techniques were introduced from the field of information sciences to soil science and geological science,so in your paper, you used box-counting method to calculate FD, which is belonged to multifractal fractal dimension, is that correct?

This manuscript is a resubmission of an earlier submission. The following is a list of the peer review reports and author responses from that submission.

Round 1

Reviewer 1 Report

The manuscript by Basirat et al. introduces an automated method for sketching the edges of a fracture network in a geological feature – cracks in a rock. While automated methods such as the Canny method are widely available, they tend to generate erroneous edges in their output, obscuring estimates of the fractal dimension (FD). The introduction of an automated edge completion algorithm in the present manuscript provides an opportunity to address these problems.

An area of strength with the present manuscript is that it does address a pressing issue for feature extraction that retains the scaling relationship in a set of fractures. This is a commendable attempt using a novel approach, and has potential for broad and significant impact with a high likelihood of interest to readers of this journal and in related disciplines (geology and image processing). However there are some issues with clarity, breadth of the literature reviewed, and the specific experiments which have been carried out

However, there are three main weaknesses or major concerns I have with the present manuscript that need to be addressed, related to the present manuscript’s findings, its place in the literature, and its potential impact.

1.     It is unclear how a report by Lee et al. of D=1.62 is comparable to D of 1.82 or 1.99 or 1.972 for the same image. This needs to be clarified – perhaps these are different images or different quantifications of fractal dimension? This underlies a much harder question. It is unclear whether the proposed edge completion method, while efficient as automated, accurately extracts edges that are present the image, or introduces edges that do not exist. This theoretical difficulty must be addressed given the disparity in D estimates provided in the paper.

2.     The manuscript would benefit from direct comparison with alternate digital image processing techniques (DIPT). An obvious one would be comparison with a Canny edge extraction – to measure against figure 4e, and its measurement of FD to compare with figure 4f. Another is the edge extraction technique proposed by Berthe and Vinoy (2011), which uses a modified Canny method that is relatively more complicated to implement, but is worth considering. Bies et al. (2016) suggest two other approaches: taking the median intensity in a grayscale image, or measuring spectral decay. An approach such as one of these would support computation of FD and is computationally more elegant/efficient than the currently proposed method. It is not clear the FD estimation from simple Canny edge extraction, Berthe & Vinoy (2011), or Bies et al. (2016) would be better or more accurate than that proposed by Basirat et al. That is exactly why comparison with such past findings is important – it would reveal the superiority of the present approach, and limitations of the alternate approaches, placing it within a related literature. I would suggest performing at least an analysis using the original Canny method and the median intensity method from Bies et al., and reporting these for comparison with the present results as a way to measure against a benchmark as well as a recent development.

3.     The second weakness is the limited scope of the references, which seem limited to books on image processing or studies of rocks. To increase the overall impact of the manuscript, it would be beneficial to consider the applicability of the technique to images other than those of rock faces and geology studies. This is true throughout the manuscript, but especially in the discussion. I would suggest adding a few citations from recent literature to the discussion to demonstrate that this question is of present concern.

Several more minor issues arise and are noted by line or figure number.

·       51 – could be reworded, “its application” seems unclear

·       54-55 – the sentence stating many studies have been conducted … should have at least a couple citations

·       63 – “the FD” should come after “geology phenomenon” for the sentence to read better

·       71-78 a few things: 1) these descriptions could be rephrased to be clearer, 2) these might be more accurate for fractals as “D approaches 0, D near 1, and D approaches 2”, and 3) while I agree that the dust image in figure 4C could range between 0 and 2, the introduction of lines connecting the points in 4C, which is what’s shown in 4E, wouldn’t the edges only be expected to range between 1 and 2? Or are they presumed to be so sparse that they may also range between 0 and 2?

·       83-93 may make more sense somewhere else in the introduction such as with the other digitial image processing work, or the other geological work, or where it is provided some rewording. I am missing why it is important that “the concept of a fractal is used in different ways in geological problems”, or whether there are other applications that are relevant (aside from the mentioned “frequency-size distribution of fractures.” As a non-geologist, I find this paragraph confusing.

·       101 – “the gray surface of the image in the point” might be better as “the gray surface height of the image at a particular pixel” – throughout the sentence/paragraph, point should be converted to pixel to be consistent with use on 102 and 103.

·       119 – “the inputs to the program written in the Matlab software” would be a better wording

·       140-183 – it would be helpful if this was split into 1) a description of the theoretical aspects (which could be moved to the introduction), including a discussion of whether the line generation process is accurate, 2) a description of the algorithm, and 3) a new section 4 of the paper – results of the present study (which would make the discussion and conclusions sections 5 and 6, respectively)

·       168-173 – the paragraph needs to be restructured because it is somewhat hard to follow. That would be an appropriate place to include a comparison with the analyses requested above.

·       191-192 – related to the confusion of 168-173, it is not clear that degree, dispersion, and/or density were determined. Please clarify here and in the preceding section. The lack of clearly stated results makes it hard to extract the pertinent information at a glance, although the degree, dispersion, and density are said to be determined here.

·       Figure 4 – could the dots in the composite in D be rendered in a color – e.g., red or yellow, rather than black, to help identify them?

·       Figure 7 – questions to answer include: 1) are box fills missing from panels a-c, 2) is that a schematic, and 3) if not, what is the rule for filling a box?

·       A final question, from my own ignorance, is whether the 2D surface image provides insights into the 3D structure of the rock – does the fractal structure of the cracks on the surface give insights into the volume. If so, it would be worthwhile to point out that this is much less intensive, computationally, and less expensive than imaging the volume – depending on the answer, this might increase the impact.

References:

·       Berthe, K.A.; Vinoy, K.J. 2011. A New Method for Segmentation using Fractal Properties of Images. International Journal of Electronics & Communication Technology, 2(4), 23-28.  

·       Bies, A.J.; Boydston, C.R.; Taylor, R.P.; Sereno, M.E. 2016. Relationship between Fractal Dimension and Spectral Scaling Decay Rate in Computer-Generated Fractals. Symmetry, 8, 66.

Reviewer 2 Report

Overall the subject matter of this paper is interesting and the paper could have merit but the study is very weak. I am mainly concerned that the entire paper is based on the analysis of a couple of images. A lot more data is required and a through comparison of the current methods to results obtained from other existing techniques must be explicitly made. The writing in the paper, especially in section 3 is very poor. The paper needs extensive revisions in terms of the presentation. I have made some comments in the attached document that must be addressed by the authors before the paper can be considered seriously for publication.   
